# Uniaxial Partitioning Strategy for Efficient Point Cloud Registration

**DOI:** 10.3390/s22082887

**Published:** 2022-04-09

**Authors:** Polycarpo Souza Neto, José Marques Soares, George André Pereira Thé

**Affiliations:** Departamento de Engenharia de Teleinformática, Universidade Federal do Ceará, Fortaleza 60455-970, Brazil; marques@ufc.br (J.M.S.); george.the@ufc.br (G.A.P.T.)

**Keywords:** 3D point cloud registration, partitioning, rigid body registration, iterative closest point, 3D surface matching, point matching algorithm

## Abstract

In 3D reconstruction applications, an important issue is the matching of point clouds corresponding to different perspectives of a particular object or scene, which is addressed by the use of variants of the Iterative Closest Point (ICP) algorithm. In this work, we introduce a cloud-partitioning strategy for improved registration and compare it to other relevant approaches by using both time and quality of pose correction. Quality is assessed from a rotation metric and also by the root mean square error (RMSE) computed over the points of the source cloud and the corresponding closest ones in the corrected target point cloud. A wide and plural set of experimentation scenarios was used to test the algorithm and assess its generalization, revealing that our cloud-partitioning approach can provide a very good match in both indoor and outdoor scenes, even when the data suffer from noisy measurements or when the data size of the source and target models differ significantly. Furthermore, in most of the scenarios analyzed, registration with the proposed technique was achieved in shorter time than those from the literature.

## 1. Introduction

Recent advances in depth sensing technology have favored the progress of research in many areas. For example, facial and expression recognition [1,2,3], robotic vision [4], UAV-pose estimation (Unmanned Aerial Vehicle) [5], rigid registration have benefited from scene information in three dimensions usually made available as point-cloud data.

In the past few years, the field of image registration has grown considerably, with the publication of new methods [6,7,8], and reviews and surveys [9,10]. In simple terms, registration in image processing refers to the mathematical operation needed to match different perspectives of a given scene by proper association of the corresponding parts present in partial views; whenever the operation assumes the form of a unique transformation accounting for rotation and translation of the entire scene, it is called rigid registration: otherwise, it is a nonrigid registration. Therefore, it is vital for scene reconstruction, which, in turn, finds numerous applications in computer graphics, human-computer interaction, robot navigation etc.

To provide an example of how important it can be, in navigation experiments of mobile robots, the fundamental problem of localization can be achieved by registering multiple views as acquired by several sensors [11], and the registration itself can be used to help determine their relative position and orientation [12].

There are several registration methods for 3D data in the literature: the Iterative Closest Point (ICP) is the pioneer among local approaches [13], and is a popular choice for efficient registration of two point clouds under a rigid transformation for its simplicity. On the other hand, the original ICP is known to be computationally demanding owing to the correspondence step, which requires up to $N^{2}$ operations for *N*-sized point clouds; it is also susceptible to the problem of falling into local minima and requires good initialization (e.g., number of iterations, threshold of convergence and initial guess, for example) to suitably prevents that issue, thus avoiding bad registration; it performs better when some data preprocessing steps are carried out, especially outlier removal, which favors the step of correspondence checks.

In line with the research focused on improving computational-effort issues, in this paper is proposed a registration algorithm relying on the ICP technique applied locally, in the reduced space of partitioned input point clouds. The main contribution of the technique is in the correspondence check, which is favored by the partitioning because it reduces the number of points used for the matching. In other words, ICP is not replaced or generalized in any sense; in fact, it undergoes a different and enhanced use when running within the proposed approach as the registration core. The introduced strategy represents a three-fold improvement of the algorithm proposed in [14], because the method is now generalized in configuration and application scope (wide range of scenarios) as well as in the spatial directions of partitioning and, more importantly, in the adopted stop criterion. Although the investigation here reported accounts on ICP as the registration core, the approach can suit other techniques and can provide them with significant improvements in the qualitative and time performance of registration.

This work is organized as follows: In Section 2 a brief review of related works is presented, which is followed by a discussion of the main differences of the proposed point-cloud-partitioning algorithm in comparison to its previous version, in Section 3. Section 4 describes the algorithm itself, and in Section 5 the materials and databases used in the various experiments as well as implementation issues are detailed. In Section 6, the results of both time performance and registration accuracy of every experiment are reported. Finally, in Section 7 a discussion is presented.

## 2. Related Works

The problem of point cloud registration is addressed in the literature from different approaches; in essence, most of the techniques either rely on methods applied to the spatial coordinates of points or on methods running on some feature space.

In traditional schools, which comprise the well-known ICP [13] and its over 400 variants (until 2011 and only those published in IEEE Xplore ©) [15], there is a lot of interest due to the simplicity and availability of solutions within open source code libraries. Among those variants, some recent implementations of this local approach deserve some attention. For instance, the Efficient Sparse ICP [16], which combines a simulated annealing search along with the standard Sparse ICP [17], which attempts to solve the registration problem through sparse-induced norms. The issue of outdoor scene registration is addressed by the authors of the Generalized ICP [18]. This algorithm exploits local planar paths at both point clouds, which leads to the concept of plane-to-plane in the registration. In that contribution, the authors generalize alignment-error metrics originally introduced in Besl and McKay [13] and Chen and Medioni [19], and although efficient, it is affected by the non-uniformity of point density over the surfaces being matched. This is interesting because it may be regarded as a dense-cloud oriented approach, but fails in non-structured scenarios or well-behaved environments, as claimed in [15,20].

In addition, the Normal Distributions Transform (3D-NDT) [21] is worthy of discussion because it makes tractable the problem of matching dense clouds and, in addition, it is a technique conceptually different to ICP; here, the scene is discretized into cells, each being modeled by a matrix representing the occurrence of linear, planar and spherical occupation of points, and then a nonlinear optimization strategy for cloud transformation is applied. However, this approach is rather time-consuming and unsuitable for low-performance hardware, as claimed by the authors in [22].

In 2014, Super4PCS was proposed [23], which solves the registration task with $O(n+k_{1}+k_{2})$, where $k_{1}$ is the number of pairs in the target cloud at a given distance *r* and $k_{2}$ is the number of candidate congruent sets. In addition, other global registration algorithms have been proposed and published. One of the most relevant is the Go-ICP [6], which combines branch-and-bound algorithms and the classical ICP.

In other schools, some important local-feature-based methods have been presented so far. In general, a feature descriptor should provide a comprehensive and unambiguous representation of geometry locally; in line with that, for instance, the relevant-based sampling approach of [24] was successfully demonstrated for point-cloud registration; good matching performance was also reported in [25], in which curvature, point density and other geometric features were employed in the correspondence step of ICP and in the error metric as well.

In general, to allow for an efficient match, descriptors are expected to have a level of robustness to external perturbations [26], or even invariance to certain transformations [27,28]. The fast point feature histogram [29] is among the most popular descriptors. The latter was used in [29] within a registration pipeline; a rough RANSAC-like pose adjustment was performed in the feature space as calculated from the FPFH, which was followed by a fine correction step with the ICP.

Despite the fact that they usually provide comprehensible descriptions of point clouds, some of the recent interest has shifted towards deep learning approaches. For instance, PointNetLK [30] is a network that can achieve matching by optimizing the distances in the feature space. Another contribution in this field is CorsNet [31], which combines the local and global characteristics of the point clouds to be matched. In general, many of the contributions are based on convolutional neural networks, which rely on several layers together with hierarchical characteristics of a large number of point cloud samples, which ultimately can limit applications [32].

In the last few years, partition or patch-based approaches have appeared in the literature. For example, in [33], the SLAM (Simultaneous Localization and Mapping) problem is addressed using a partition-based approach responsible for finding a number of tagged objects, making them useful for scene registration. Instead of objects-of-interest, Fernandez-Moral et al. [34], performed segmentation to look for plane surfaces in 3D scenes prior to registration.

This way of approaching the registration of an entire scene from a small part of it is interesting *per se*, because it means saving computation efforts; the drawback is that, if no proper care is taken, the surface under interest might suffer from significant loss of information and/or inevitably high ambiguity, what would ultimately limit the ability to retrieve orientation; another negative side is that the inclusion of tagged objects to favor segmentation could be argued as a non-acceptable intrusion in the scene for a given application.

A way to circumvent this problem is to consider small parts yet spanning the entire scene. This was the rationale of the cloud-partitioning ICP (CP-ICP) introduced in [14], in which the source and target information of the object to be registered are sampled along a given spatial direction into slices of point-clouds; in different iterations, only the slices undergo pose correction by ICP. The method was improved in [7] by the inclusion of a sufficient matching stop criterion. The technique was further improved to provide flexibility in the spatial direction of point-cloud sampling among the three principal axes the local frame and tuning of the stop criterion, giving rise to the version presented in this manuscript, named Uniaxial Partitioning Strategy (UPS), which will be detailed in the following pages. It should be emphasized that the approach can accommodate different variants according to the technique implemented at the registration core, hence, it can find the interest of researchers developing the field of point cloud registration.

## 3. Our Contributions

In this new approach to point cloud registration, three major improvements were made compared to [7]:(a)The partitioning approach is now applicable for retrieving orientation resulting from multiple rotation phenomena around general axes; in addition, there is now flexibility in the principal axis along which cut-sectioning is done. Compared to previous versions, the choice of the axis is now automatic. Currently, the algorithm chooses the cutting axis after measuring the data variance along the three principal axes of the local frame. For more details, see Section 4.2.1;(b)The method now has two operating modes, namely configurations A and B, which refer to the chosen cutting axes; they can either be different (configuration A) or the same for both target and source clouds (configuration B). Configuration A allows partitioning source and target models in different directions, what sounds useful for registration where point clouds come from different acquisition systems, for example.(c)For that which concerns the stop criterion, it is now calculated for every input cloud on the basis of an original proposal called micromisalignment (detailed in Section 4.2.3), which is conducted automatically, implying no need for previous ad hoc knowledge of the input models. To the best of the authors’ knowledge, no other work in the recent literature suggests a measurement for registration goodness based on the input model itself and, as such, automatically adjustable. Other approaches instead rely on the use of parameters or constants of limited scope.

Comparing to the literature, it is to be emphasized that our technique is a geometry-preserving approach, since it works on the full ensemble of points, contrary to some sampling techniques relying on representative points that do not belong to the data itself, or even to descriptor-based methods that work in a space other than the original data, and yet ours performs well in terms of both time and quality of alignment aspects. In addition, the results reveal that the applicability extent of UPS is demonstrated for models ranging from simple rigid objects to more interesting indoor and outdoor scenarios. Finally, robustness to noise corruption was also assessed.

## 4. Uniaxial Partitioning Strategy

Consider two surfaces represented as point clouds: a source and a target point cloud. The problem addressed is to successfully match them in position and orientation, that is, find the rigid transformation representing the best overlap. Originally, this problem was solved by the ICP algorithm, which is located in the nucleus of the UPS algorithm in the present investigation. Nevertheless, other approaches may occupy the central part of it, such as ICP point-to-plane and Generalized ICP. New joint strategies combining the partitioning proposed here and deep learning approaches can be evaluated, in the future, if the space of subclouds is used for data augmentation.

The algorithm starts by dividing each input cloud (source and target) into subclouds. Then, these subclouds are iteratively subjected to pairwise ICP registration, which results in an orientation matrix. Pose correction of the original (entire) point clouds is then assessed from the matrix obtained and from a quantity check, which works as a stop criterion. These steps are discussed further in the following subsections.

### 4.1. A Look at ICP

Before describing the method in detail, it is worth mentioning that the partitioning affects ICP in a three-fold manner: selection of points, matching and metrics for alignment check.

#### 4.1.1. Selection of Points

It is a good practice to run a pre-selection of points, thus reducing the effort in the correspondence check step. In this regard, the literature introduces many strategies, such as random [35] or uniform downsampling [36,37]. Although classical sampling strategies may be very useful for computational reasons when dense clouds are concerned, sparse point clouds might lead to the loss of relevant information. In addition, in the partitioning approach there is a selection of points (because the cloud is not presented entirely for ICP pose correction), but in the form of small pieces at a time, meaning that the cloud can occasionally be assessed entirely whenever the iterations over the partitions fail to meet the alignment requirements.

#### 4.1.2. Matching

The correspondence check of the ICP is the step responsible for proper point-association between every point of the target cloud and every one belonging to the source counterpart. It is very time-consuming, and in the literature, it is favored by known approaches for data structuring, such as kd-tree [38]. In the present implementation of UPS it undergoes the same way, but in the reduced space of subclouds, which ultimately reduces the *N* × *N* search for correspondence step. It is worth mentioning also that here it was used the Point Cloud Library [39] implementation of ICP, with built-in call to kd-tree search, although it may be replaced by other alignments algorithms, according to the advances in the state-of-the-art and to the application specifications.

#### 4.1.3. Error Metrics

Among the known choices for error metrics to be minimized, classical point-to-point metrics were adopted. However, this method relies on the concept of micromisalignment of the entire input cloud, which can be tuned more or less strict according to the specific aspects of the considered scenario. More details regarding this concept are provided in the upcoming sections.

### 4.2. Mathematical Formulation

In this new approach for point cloud registration, partitioning and the stop criterion play central roles. Here, the underlying mathematical formulation is presented.

#### 4.2.1. Partitioning

Given two input models, namely the source (*S*) and the target (*T*) models, respectively, $N_{S}$-sized and $N_{T}$-sized point-clouds, partitioning is the operation of grouping the points comprised by each of them into smaller sets, amounting to k−1 groups, hereafter called subclouds and indexed by letter *j*. The grouping occurs by means of a cross-sectioning of the given cloud and *k* planes along a principal axis, ξ-axis (which can be *x*, *y* or *z*-axis). This is illustrated in Figure 1 for the Dragon model cut along the *y*-axis.

Partitioning can be described as follows. Let Equations (1) and (2).
(1)$$S=\left\{s_{i}\right\},\;i\in N,1\leq i\leq N_{S},\;$$
(2)$$T=\left\{t_{m}\right\},\;m\in N,1\leq m\leq N_{T},\;$$
is the original source and target models and let Equations (3) and (4)
(3)Sξo=sio,i∈N,1≤i≤NS,
(4)Tξo=tmo,m∈N,1≤m≤NT,
is the corresponding sets after ordering along the ξ-axis in such a way that the above equations respective of the following, Equations (5) and (6):(5)sio·ξ^<si+1o·ξ^,∀i,
(6)tmo·ξ^<tm+1o·ξ^,∀m.

Finally, the subclouds are then created as $\frac{N_{S}}{k}$ and $\frac{N_{T}}{k}$ -thick slices of the ordered point clouds, according to the following, in Equation (7) and (8):(7)Sj={sio∣(j−1)·NSk<i<j·NSk},
(8)Tj={tmo∣(j−1)·NTk<m<j·NTk}.

As should be noted, the partition-axis must be chosen before the grouping itself; in this regard, UPS offers two variants, namely, Configurations A and B. In both variants, the cutting axis is chosen after calculating, as a measure of dispersion, the variance in the principal axes of the local frame. The choice of sectioning along the *x*-, *y*-, or *z*-axis is in part related to the fact that the data are organized as a list of point coordinates in point cloud files. Among the three alternatives, the direction of the maximum data dispersion is considered as a criterion guiding the choice of the cutting axis.

The reader could think of determining the cutting axis by using PCA (Principal Component Analysis), since it provides an insight on data spatial dispersion. The point is that its outcome could suggest any spatial direction and not only those corresponding to the three principal axes of the local frame of the acquired point cloud and that would come at a cost. Since UPS relies on the simplicity of cutting along *x*- or *y*- or *z*-axis and nothing else in its conception, only the essentials of PCA were inherited and kept, namely, the computation of variance along the main axes mentioned above.

That maximum data dispersion criterion for the cutting axis is also a choice coming from observation and aims at favoring a reasonable tradeoff between the average slice size (in number of points) and the number of slices. In this regard, it was found [14] that these are antagonic aspects; they affect rapidity and registration quality in opposite ways. In that sense, the axis along which data spreads more allows for a larger number of slices, meaning more flexibility when tuning this important algorithm configuration parameter. It is not by any means assumed that the given surface has the maximum data dispersion along one of the three principal axes. Indeed, in the general case, data can be maximally distributed over any arbitrary oriented direction. Hence, it is speculated that cutting along one principal axis suffices for the purpose of finding good candidates for registration among subclouds.

From the comments above, configurations can be synthetized as follows:Configuration A: In this configuration, the partition-axis of a given input model is chosen as the one with the largest data variance among the three principal axes. Therefore, source and target models can be cut along different ξ-axes, which might benefit scenarios in which they differ significantly in orientation (for example, where clouds are randomly rotated [6] or captured by different sensors [40]).Configuration B: Here, data variance is calculated only in the target point cloud and the chosen ξ-axis is assigned to both input models, performing faster than the previous one for obvious reasons. It might be a good choice for situations in which the ground truth is known, as well as for registration of sequentially acquired shots in which orientation changes only in one degree of freedom (for example, in-plane robot navigation in SLAM applications [41]).

#### 4.2.2. Convergence Check

As mentioned earlier, the correspondence check step is performed here in the reduced-space of subclouds; hence, the objective function to be minimized is slightly modified to address the fact that there are iterations over subclouds in this partition-like approach. In simple words, compared to the cost function of the classical ICP, it appears to be dependent on the subcloud index, *j*, see in the Equation (9):(9)$$F_{j}\left(q\right)=\frac{k}{N_{S}}\sum_{i=1}^{\frac{N_{S}}{k}}\left\|t_{m}-\left(R_{j}\left(q_{R}\right)s_{i}+q_{T}\right)\right\|,$$
which is defined for j=1,...,k. This implies that the index range of points $s_{i}$ and $t_{i}$ entering the summation covers the ensemble of points lying in the *j*th subcloud.

In addition, in the cost function, the parameters $q_{R}$ and $q_{T}$ are, respectively, the orientation vector represented by a unit quaternion and the translation vector; the former can be used to give the rotation matrix $R_{j}$ appearing in the Equation (for details, see [42]). These vectors can be put together in compact form $q={\left[q_{R}|q_{T}\right]}^{t}$, thus allowing the retrieval of the complete registration vector relating subclouds $S_{j}$ and $T_{j}$, in the same way as described in [13], adapted to the reduced space of subclouds.

Finally, it should be emphasized that the registration vector (and, consequently, the rotation matrix) is obtained every time a pair of subclouds is subjected to matching, which is then extended to the full space of point clouds through the transformation of the input target cloud by the matrix $R_{j}$ and posterior quality checking of alignment to the source cloud. The misalignment between the target and source clouds is calculated and compared to φ, a threshold value that represents the stop criterion. The details of our proposition for such a threshold value are given next.

#### 4.2.3. Stop Criterium

Our proposed method for calculating the threshold value is very simple. Initially to φ it was assigned the deviation as measured from the root-mean-square error between the target point cloud and a slightly 3D-misaligned copy of it. A slight misalignment indicates a change in orientation around the three degrees of freedom. This is called micromisalignment hereafter.

The concept is illustrated in Figure 2. Essentially, from a frame *A*, representing the original pose of the input model, a micro-misaligned frame *B* can be obtained if small rotations are applied successively around the principal axes. In the example illustrated, it starts with a rotation around *Z* by an angle α, thus giving rise to intermediate frame $B^{\prime }$, then around *Y* by β, giving rise to intermediate frame $B^{\prime \prime }$, and finally around *X* by γ, leading to the final frame *B*.

Since these are rotations around local axis, the Z-Y-X representation for the Euler angles can be used and, imposing the rotation angles around the three axes to be equal amount, α=β=γ, the resulting rotation matrix resembles such as the following, in Equation (10):(10)RZYXBA=RZ(α)RY(β)RX(γ)=Rθ=cθ2cθsθ2−sθcθsθ2cθ+sθ2cθsθcθ2+sθ3cθsθ2−cθsθ−sθcθsθcθ2.
where $c_{\theta }$ and $s_{\theta }$ are short forms for cosine and sine of θ.

Once the rotation matrix of the misalignment is multiplied by the input target point cloud, a slightly misaligned point cloud is obtained, that is, a micro-misaligned instance of it is available, from which it can be calculated a deviation measurement quantifying the threshold value for the stop criterion, in Equation (11):(11)$$\varphi =\sqrt{\frac{1}{N_{T}}\sum_{m=1}^{N_{T}}\left\|t_{m}-R_{\theta }t_{m}\right\|},$$
where the summation spans over the entire ensemble of points of the target cloud. As mentioned so far, this quantity represents a lower-bound to be reached by the root-mean-square error between the input target and source clouds. The reader should realize that the smaller θ is, the better the misaligned copy fits the original cloud, meaning that the RMSE (in meters) between the two instances of the cloud expresses a quantitative measure of good matching, and thus can be used as a stop criterion threshold in the algorithm. In that measurement, RMSE is calculated as the root mean square of the 3D distance between each point of the target cloud and its corresponding closest one on the micromisaligned target cloud. More details on this parameter are provided later in this paper. For completeness, the UPS algorithm flowchart is shown in Figure 3.

## 5. Materials and Methods

In this section, the methodology is introduced, describing the datasets used in the experiments, as well as the occasional modifications performed in the point cloud models for augmenting the set of investigation scenarios.

Our research was organized into seven experiments; from Experiments A to G, the aim was to progressively add complexity to the investigation. In doing this, the analysis started with pairwise registration from simple models for which the ground truth is known, and then we moved to more interesting situations in which the number of partial views increases, or the models suffer multiple rotations, or the models suffer from noise addition, or yet they correspond to outdoor scenes, etc.

In the various experiments, the registration algorithms investigated changed accordingly, in such a way to favor fair comparison to the literature or because in some experiments, the use of some techniques simply would not make sense. The list of the algorithms used in this investigation comprises Go-ICP and Go-ICP Trimming [6,43], Sparse ICP [17], FPFH approach [29], CP-ICP [7] and ICP variants [13,18,19], 3D-NDT [21] and downsampling methods [35,44,45].

In addition to the models made available with PCL [39], the models were obtained from different sources: Parma University models [46], Statue Model Repository [17,47], Stanford 3D Repository [48], as well as the particular data used in [23], indoor scenes homemade acquired in our laboratory from Intel RealSense SR 300 [49], scenes acquired outdoors on our university campus using the SICK lidar sensor, and ASL datasets [11].

For more details on the point clouds used in the Section 6, Table 1, which is devided into the categories objects and scenarios, brings some characteristics, such as density (approximate number of points ) and average file size. In order not to be repetitive, it is worth mentioning that all files have the *“.pcd"* extension (Point Cloud Data).

In addition to qualitative assessment (from simple visualization) of the various point cloud matching performed so far, precious quantitative information is provided; such analysis relies on four main quantities:the running time (in seconds) of the registration as required by the algorithms implemented;the RMSE measure (in meters; here computed between the source cloud and the target cloud after pose correction);the estimated pose calculated according to the equivalent angle-axis representation for orientation;the mean RMSE, calculated as an average over the 3D models used in each experiment.

### Implementation Details

To allow for future validation of the proposed technique, a few notes regarding its main configuration parameters are worth noting. The misalignment, θ, gives the algorithm the ability to set the quality of pose correction as flexible as the application requires; they were analyzed, in preliminary tests on Happy Buddha, Dragon, Horse and Hammer models the effects of making the criterion as rigid as θ = 0.1${}^{\circ }$, as well as its route towards larger values until θ = 3.0${}^{\circ }$. For those models, Figure 4 plots the trend between the micromisalignment angle and the corresponding RMSE calculated from the cloud and its micro misaligned copy. The picture also shows, with circle markers, the RMSE calculated from source and target clouds when the ground truth is applied for pose correction. One can see that θ = 2.5${}^{\circ }$ is the lower limit for the entire set of models and is imposed by the Hammer model. Trying to force an even shorter angle would make the algorithm to get stuck, since not even the pose alignment according to the ground truth is able to give the corresponding small RMSE. In other words, it gives the closest RMSE to the ground truth achievable by an intentional micromisalignment.

Another important configuration parameter, the amount of subclouds, *k* represents a quantity for balancing the time performance and the amount of data needed in a subcloud to achieve a good global matching. In this paper, upper and lower limits were defined to 1000 points/subcloud and 2000 points/subcloud, to guarantee that sparse and dense models belonging to a wide set of experiments could be registered with minimal user action. Finally, to determine the ICP convergence (which is by the way at the registration nucleus of our method), the maximum number of iterations was chosen to be 30 trials; this is in accordance with the range (30–50) suggested in [13], and to preliminary investigation made in the Bunny and Dragon models.

To make a fair comparison throughout the entire set of experiments, the ICP parameters were kept equivalent for all the variants reported here, and the various techniques followed a preliminary study for parameter setting from information made available by the respective authors. Most of the algorithms used in the scenarios were written in C++ in the framework of the PCL library [39], with the exception of Sparse ICP and Go-ICP (and its trimming variant), whose executable versions were made available after [6,17], respectively. The platform used was Intel^®^ Core™ i5 and 8 GB RAM. To see the programs of the techniques present in the Section 6, access the following link https://github.com/pneto29/UPS_paper accessed on 15 February 2022). For details on parameters, order of parameters and suggested values, see the README.md files in each subdirectory.

## 6. Results

### 6.1. Simple Pairwise Registration

This experiment is aimed at comparing the various algorithms at pairwise registration tasks, in which two views of different objects are subject to registration by Go-ICP, Sparse ICP (as provided does not output the retrieved rotation), classical ICP, CP-ICP, and the UPS. It should be emphasized that some of those techniques were evaluated under different configurations (for example, Sparse ICP with 30 or 100 iterations, UPS with flexible choice of partitioning axis, etc). Quantitative assessment is based on RMSE as well as on the rotation as given by the equivalent angle-axis representation, since the ground truth was available for the 3D models in this experiment.

Table 2 shows the performance of each technique as running time in seconds. In boldface we highlight the fastest technique for each 3D model evaluated. According to it, the UPS techniques achieved good performance, except for the Horse model.

Additionally, the rotation obtained for each model was analyzed; this is summarized in Table 3. In bold we highlight the technique that is closest to the ground truth. It can be observed that UPS achieved superior performance as it approached better the ground truth for most of the models.

A comment is worth making regarding the choice of configurations A and B of the UPS, as Table 3 and Table 4 reveal, both configurations achieve the same pose correction performance, which is a very simple experiment in which the rotation phenomenon is around a single principal axis (*z*-axis in the present case), a typical in-plane robot navigation system, for instance, could, in principle, benefit a lot from configuration B, since in that application the change of orientation is usually uniaxial.

It is to be mentioned, however, that the quality of pose correction as measured by the RMSE was globally superior, even for the Horse model, as it can be checked from Table 4, which brings this measure for the various methods and models.See the values in bold, they represent the smallest RMSE values.

For the sake of illustration, the registration of Bunny and Horse models are depicted in Figure 5a–d for different methods. To be highlighted is the CP-ICP limitation when performing the registration of Bunny model; since this is a *Y*-axis rotation, and the CP-ICP is limited to partitioning along the *Z*-axis, the overlapping between subclouds of source and target is affected.

Finally, the flexibility of the UPS to find a good axis for partitioning was evidenced in this initial experiment. It can be seen how counterparts fail, especially when facing the Horse model, which was used to impose the hard initial condition of large orientation deviation (180${}^{\circ }$). This might confirm the expectation that classical ICP better suits fine pose correction. Furthermore, the results of Figure 5b reveal that our approach enhances ICP performance by giving it the ability to successed in coarse pose correction.

### 6.2. Registration under Combinations of Arbitrary Rotations

The goal of this experiment is to see the ability to match the views of the source and target when successive and arbitrary rotation phenomena around the principal axes occur; hence, no ground truth is given. The algorithms used for comparison are Go-ICP (which claims to be able to deal with such a scenario), as well as Sparse ICP. See the results in Table 5, where the UPS performs better than the others.

In Figure 6a,c, the initial poses of the two models are shown and reveal how different they are (in position and orientation) after several arbitrary transformations intentionally carried out. This is the case of an unknown rotation about an arbitrarily oriented axis, and not about some of the principal axes. Once again, for illustration we bring in Figure 6b,d the final pose as obtained after registration.

At this point, UPS’s ability to retrieve aggressive rotations as those reported in Figure 6a–d can be questioned in case of shapes that generate a subcloud space of ambiguous surfaces. For instance, cylindrical-shaped objects such as the Hammmer model of Figure 7. That figure reports the result of UPS registration for a scenario in which target and source model deviate to each other by two successive rotations of 90${}^{\circ }$ degrees around two principal axes. Although the two clouds have low overlap in the universal frame space, the algorithm was able to solve that case because it has the flexibility to choose different cutting-axes. This way, it can generate a subcloud space of slices which do have correspondence among themselves. Here, the algorithm worked in Configuration A, cutting along the *Y*-axis the source model and the *Z*-axis the target model with 4 slices of nearly 500 points each. After 1.05664 seconds, the algorithm satisfied the stop criterion quality and reached RMSE=0.00407211.

### 6.3. Downsampling Effect

Until now, it has been stated that our proposal is not a sampling, but rather a patching-like approach for registration, and the more evident benefit of it is the reduction in time effort achieved with the cloud slicing already described. Nonetheless, one could argue that it is quite similar to the concept of sampling, since every time the registration nucleus of the algorithm comes to the scene, it operates not in the full ensemble of points of the given cloud, but in a small portion of it instead. To cope with this plausible understanding, in this section it is studied how UPS performance is compared to known sampling procedures usually applied along with ICP. Uniform sampling was chosen, with 67% size reduction for Buddha and 45% for Dragon, as well as random sampling with 50% and 70% size reduction. Once again, the results are summarized for the time and quality of pose correction, and are shown in Table 6 for both the Dragon and Buddha models. See the values in bold that represent the best results.

The numbers in the table reveal that ICP aided by uniform sampling was faster than our method, but it came at the expense of bad orientation correction; indeed, Dragon and Buddha ground truths were better approached by UPS, with acceptable time-performance especially in configuration B. To provide support for these numbers, the visualization of Buddha registration in Figure 8a–c shows evident misalignment after ICP aided by sampling methods.

### 6.4. Registration in the Presence of Different Levels of Gaussian Noise

In this section, the existence of additive Gaussian noise in both source and target clouds was investigated, emulating the effect of measurement uncertainties associated with acquisition issues. The independent parameter is the standard deviation of the *z*-coordinate, which lies whitin the range of (0.002–0.005). Here, the investigated methods are Go-ICP and its Trimming version (with parameter ρ set to 0.1), the ICP variant based on FPFH, Sparse-ICP (with iteration limits set to 30 and 100), and ours. The model considered was the Bunny point cloud. Table 7 shows the running time in the registration for the different conditions of noise intensity, revealing that the UPS in configuration B outperformed in this experiment and was able to beat the other methods, being at least two times faster in most scenarios. Best results are boldfaced for better comparison.

Table 8 instead shows the quality of pose correction as measured by RMSE. Some remarks can be drawn here: (a) ICP based on FPFH descriptor and the Sparse variant performed poorly; (b) Go-ICP and the Trimming version also showed bad performance, what might be associated to the inherent sampling step needed for optimization issues in those techniques, and the existence of increasing uncertainty in data degrades the surface representation provided by sampling. (c). UPS is not affected in terms of the quality of pose correction, and the time performance started suffering only at a large noise intensity level. (d). Beacause all these techniques are subject to the point-correspondence step of ICP, the results suggest that they performed differently because they act differently in surface representation and because they differ in their intrinsic ability to overcome the bad representation caused by noise addition. In that view, our approach suffered less because it does not change the surface representation by any means, it breaks the surface into small pieces. Best results are boldfaced for better comparison.

To illustrate the performance of UPS compared to the other methods, the registration of the Bunny model for σ=0.005 was shown in Figure 9a–d.

### 6.5. Partial Registration of Point Clouds with Different Overlap Rates

In many situations of 3D scene perception, including cluttering and occlusion scenarios, the models which go into registration do not present full overlap. It is a rather frequent concern, hence, the ability of the registration algorithm to deal with models presenting partial overlap of common regions in the target and source surfaces. Since ours is a partitioning approach, one could argue that the partial overlap scenario can be particularly challenging for it. In this section, this issue was addressed on the basis of an approach similar to [50]. Our goal is to investigate the extension of the method towards the partial overlap of the objects studied so far.

Let us start with the dragon, bunny and buddha models at overlap ratios of 25%, 30%, 50% and 75% between source and target data. Results are shown in Figure 10 along with the ground-truth (named as $\theta_{before}$) and the retrieved orientation obtained by UPS (named as $\theta_{after}$). The reader can see that UPS performed well for dragon and buddha models, but did not for the bunny model at low overlap ratios. The observed decrease in the alignment quality is associated with the lack of surface representation in the generated slices after our partitioning approach, meaning that the correspondence step of the ICP algorithm running in the inner level of the method starts failing at low overlap ratios. Nonetheless, it is worth emphasizing that the bunny model poses challenges due to the larger orientation deviation between source and target shots (of about 45${}^{\circ }$) compared to the other models, which amounted to 24${}^{\circ }$ instead.

In general, according to the results, higher overlap ratios obviously lead to good source-target matching, with retrieved orientation approaching the ground-truth. Although that conclusion was expected already, it is useful to point to a second issue to address: the region of overlap itself, and its hypothetical influence on the alignment quality. We have been pursuing so far (since the *début* of our partitioning proposal in [14]) that there are preferential regions for the matching and that the exhaustive search for it among the slices of points is just one way to find them.

In line with this, the study was moved to the situation of 50% overlap ratio as before, but this time changed the portion of the models overlapping to an intermediate region. Figure 11 helps us to understand that, for the dragon, it means that the feet and the head do not undergo registration, whereas for the bunny, the upper part of head as well as the ears are neglected along with the feet. The reader should notice what a good registration was achieved for those models. On the contrary, for what concerns the buddha model, the use of the intermediate portion led to a challenging situation and UPS was able to retrieve the orientation with about 10${}^{\circ }$ error (see inside the red rectangle how the feet of target and source deviate to each other).

To better comprehend this, a look at the slices $T_{j}$ and $S_{j}$ which “won” the registration according to UPS can be useful. Those are plotted in red colour in the rightmost pictures of buddha model at the bottom part of the Figure 11. The picture also highlights the centroids of both target and source winning slices marked by tiny blue circles. The bad registration in this case can be explained by the distance between those centroid points, what leads to unreliable translation at the initial steps of the ICP algorithm. Hence, for better use of the partitioning approach it is strongly suggested the input parameters to be set to provide similar amounts of points in the target and source slices.

Partitioning is splitting source and destination into k subclouds, aligning source and destination subcloud pair with the same index. In the condition where we have similar density/resolution, aligning sub-clouds of the same index allows us to align topologically corresponding slices.

In situations where there is a density difference, one of the clouds corresponds to a tiny portion of the other, the models will be partitioned into different numbers of sub-clouds of similar size; however, the indices that will be aligned do not present topological similarity. Aligning sub-clouds arising from partitioning between disproportionate original clouds, with low overlap, leads to poor alignment, as can be seen in the example of the Buddha model, in Figure 11.

Obviously, future versions of UPS may consider partitioning into different numbers of subclouds and also consider associating different pairs of source and target slices (for example $T_{j}$ with $S_{j+1}$ or $S_{j-1}$). This could lead to a more general search space of candidates to match and, as such, better outcomes could be achieved. In that augmented space, one interesting initial investigation to point attention is on close (but non-contiguous) sub-clouds, since the proximity helps preserve the correspondence of surface points. Our guess is, however, that would come at the cost of rising computing efforts, and that is why it has not been considered so far.

The results of a final investigation regarding the partial overlap scenario scenario are shown in Figure 12. Here, we were interested in taking the worst alignment case as measured by the retrieved orientation obtained by UPS and comparing it to other approaches from the literature. This was conducted for the buddha (75% overlap, $\theta_{after}$ = 23.1${}^{\circ }$) and bunny (25% overlap, $\theta_{after}$ = 38.9${}^{\circ }$) models. The approaches chosen were Go-ICP and Four Point Congruent Sets (4PCS) [51] because these are claimed as good candidates for partial overlap registration scenario and appear in the wide set of techniques investigated in the present work. Middle and right columns of the picture show that those approaches did not beat the quality of registration achieved by UPS.

### 6.6. Registration of Indoor Scenes

In this section, in this section, the investigation context changes from single objects to scene registration, starting with indoor environments and registration in pairs. In addition to using homemade acquisitions in our lab (named Lab. 1 and Lab. 2 models), the Office and Stage models from [23] were also considered. The time performance of the various methods is summarized in Table 9. A look at the numbers reveal that UPS was less efficient in the registration of Lab. 1. Notice the best results, highlighted in bold.

Nevertheless, when it comes to quality of pose correction, as measured by the RMSE (see Table 10), once again the UPS shows good performance, somehow compensating for the lack of time efficiency. See lower RMSE values in bold.

To illustrate a case of indoor scene registration, in Figure 13a,b the shots of Lab. 1 model is shown after the registration. The inaccurate estimate of the 3D-NDT algorithm negatively affects the registration result, whereas the proposed algorithm achieves satisfactory alignment.

### 6.7. Registration of Multiple Shots of Indoor Scenes

Multiple views of the indoor scene were then considered for cascade registration; here, the models used were Lab. 1, Lab. 2 and House, and the registration methods were kept the same as those of the previous experiment. Three different shots are available for each model. As expected, the increase in the number of shots to undergo registration is reflected in the time performance, as listed in Table 11. Nevertheless, it is to be stressed that once again UPS was superior in terms of time performance and achieved good quality of pose correction, as shown in Table 11. Because the different views have significant overlap, the results suggest that the other approaches do not take profit as much as UPS does. Note the bold markings in the table below, they indicate the best results for each metric.

### 6.8. Registration of Outdoor Scenes with Different Point Densities

This extensive study ended with a scenario particularly useful for instrumentalists, in which it was considered very different spatial sampling rate, leading to source and target clouds with different data densities, which is usually the case when acquisition comes from different sensors. Hence, the task to be accomplished is to register sparse and dense clouds corresponding to different shots of a given outdoor scene. For this assessment, point clouds were chosen from the Gazebo Summer and UFC datasets, as provided after [11] and homemade acquisitions, and once again single registration between two views was performed.

It should be remembered that the target cloud is not a sampled version of the source cloud at all; if this was the case, one could argue that it would be nonsense to evaluate ICP-based techniques, since the step of correspondence check could easily (except for trapping in local minima) give perfect matching for the entire ensemble of points of the smaller set.

As it has been conducted so far, the time and quality of the pose correction are reported in Table 12. It is worth mentioning the huge size of one of the shots from UFC dataset, which amounts to nearly 1.2 million points; the other shot is about 828 k points in size. Regarding running registration time, the results reveal that both configurations of UPS performed better than their counterparts. This is also the case for the RMSE metrics.Note the bold markings in the table below, they indicate the best results for each metric.

Globally, this last experiment suggests that UPS could be a choice for embedded solutions even for outdoor 3D mapping from heterogeneous acquisition setups. For completeness, the registration for UFC dataset is depicted in Figure 14b.

Although the approach performed very well in a number of challenging scenarios, a final investigation is worth giving an idea on its limits and when it fails. We stressed the experiment on outdoor scenarios to cope with the case of low overlap between models and significantly different densities. Here, point clouds acquired after UAV flights over a coal stockpile in a thermal power plant were evaluated. As illustrated in Figure 15a, the source and target models are very different in amount of points and so it is the subcloud size of the slices undergoing registration. As it can be seen in Figure 15b, UPS was not able to retrieve the transformation, though it was mostly due to translation. This points to some limitations regarding the difference in point clouds densities, which affects the partitions and their ability to keep topological information. This directly affects the matching core; indeed, point-to-point fails probably because it has centroids estimation and translation recover as initial steps and, for clouds placed closely as in this case, falling into local minima is likely. As a matter of fact, this last issue could motivate the use of automatic selection of alignment core within the proposed partitioning approach.

## 7. Discussion

For completeness and better comprehension of the extensive investigation of UPS, a number of comments shall be made.

In comparison to the traditional ICP as well as to its variants [13,19] assessed in this work, the improvement achieved in registration quality may accept two explanations: first, the space of subclouds can offer better alignment conditions than the original clouds; in addition to that, the existence of a micromisaligned-tuned RMSE to be met helps preventing the ICP convergence to local minima. With regard to the computation efforts, the performance achieved is associated with the fact that UPS operates in reduced-size space, which was confirmed from the running time in the entire set of experiments.

In its classical implementation, ICP is dependent of an initial rotation guess, which is an important limitation of the algorithm. On the contrary, the UPS approach does not depend on that. To circumvent the mentioned limitation, some techniques rely on coarse registration. As a matter of fact, the use of FPFH or even its use along with RANSAC have been reported in the literature. Nevertheless, those solutions impose limitations on the amount of relevant features necessary for proper surface description, as well as on the conditions for convergence, afterall increasing the computational efforts [52,53]. This is observed in Table 3, with FPFH + ICP${}_{p2pt}$ outcome of 50.154${}^{\circ }$, which represents a large deviation from the 24${}^{\circ }$ ground-truth, whereas UPS pose correction amounts to 24.039${}^{\circ }$. In addition, Table 2 brings important numbers in favour of UPS; for example, it reveals that, for the Buddha model, UPS${}_{cf.B}$ performs up to 171.131 times faster than FPFH+ICP${}_{p2pt}$.

Versatility is also an important feature of the algorithm. Starting with the micromisalignment concept introduced here, it can be claimed that it simplifies the need for a RMSE-based stop criterion, making it easier to tune for different scenes. In addition, it represents an interesting adjustment resource for making the matching more or less strict, as imposed by the application or by the computational resources available. It can be mentioned that this concept can be applied to quality measures of pose correction other than RMSE as the scientific community advances in this still open issue.

The existence of two operation modes also indicates its versatility. It must be said that it favors adaptability to different applications. Indeed, partitioning along different axes of source and target models (configuration A) allows for proper and automatic treatment of scene registration in non-controlled environments where little or no previous information is available. On the other hand, the use of configuration B favors scenarios in which data acquisition of target and source point clouds differ roughly by only one degree of freedom, thus making the most of its time performance.

The problem of partial registration is an important issue in the field, and UPS was able to solve it to a good extent. In the literature, the existence of occlusion or self-occlusion has been approached in different ways, as stated in [53], including the Four Point Congruent Sets (4PCS) [51] and the recent proposition of Wang et al. [50] provided important techniques to deal with that. In the investigation conducted here, 4PCS did not perform well (see Figure 12) and, in addition to that, it requires the use of an overlapping ratio parameter. Conerning the study presented by Wang et al. [50] on partial registration, good results for the overlapping ratios of 25.18%, 27.39% and 31.67% were reported. In our investigation of the overlapping ratio changing from 25% to 75% applied to the same models as in [50], UPS also performed well. Moreover, in the present study, the more challenging situation of input clouds corresponding to shots acquired from different perspectives was considered.

Concerning the registration of outdoor scenes, UPS performance was compared to usual choices for that scope, which includes ICP point-to-plane [19], Generalized ICP [18] and 3D-NDT [21]. These algorithms faced difficulties because the input models comprised unbalanced shots (with point clouds different in size). From a quantitative perspective, 3D-NDT showed limitations before the use of inputs from the UFC dataset, and the goodness of alignment as measured by RMSE was about 16.165 times worse than that achieved by UPS (see Table 12). Similar performance was found for Generalized ICP. Concerning elapsed time, UPS${}_{cf.B}$ reached good registration sooner than 3D-NDT (4.205 times slower) and much sooner than Generalized ICP (48.946 times slower).

In the same context of point clouds having different size, Tazir et al. [40] introduce the concept of cluster-ICP registration by a normal-based selection of surface regions and compare it to NDT, GICP and ICP${}_{p2pl}$. According to the results shown therein, CICP reaches convergence shortly, with nearly half the iterations required by ICP${}_{p2pl}$. Instead of relying on the calculation of normals (which is costy for dense clouds [54]) and making an a priori selection of local regions good enough for global registration, UPS assumes that knowing in advance the best region is not a must. Indeed, amongst the subclouds generated after the partitioning procedure, anyone can be a good candidate; UPS then lookup for it as long as the quality criterion is not met. The simplicity of such an a posteriori check of goodness revealed to be more than enough for achieving remarkable matching efficiency in several scenarios. Comparing it to the same algorithms as in [40] for the UFC dataset pointclouds, UPS performed at least 4 times faster. From the above mentioned, UPS is simpler and does at least as good as CICP.

To provide a view-at-a-glance about the comparison to the literature, in Table 13 we list some of the issues mentioned as well as some other important aspects to consider in point cloud registration along with the ability of each algorithm to deal with them. In the table, “×” means that it is not satisfied and “✓” means that that criterion fits the application.

Continuing the discussion, now about the proposal of this work, it should be emphasized that the use of partitions itself is a concept and, as such, it can be adapted to other registration *nuclei*; it is therefore left for future work the adoption of techniques other than ICP. Furthermore, the procedure for the selection and composition of subclouds may be an object of future investigation.

Indeed, as presented here, UPS creates partitions obtained after sectioning the point clouds into contiguous slices of a given axis, with each of them being considered (in pairs, $S_{j}$ and $T_{k}$, and sequentially, j=k=1,2,3...) as inputs to the UPS core. Although this simple configuration was able to reach and overcome the performance of most algorithms assessed, other choices concerning the way the pairs $S_{j}$ and $T_{k}$ are taken can be investigated. In other words, it is to be analyzed the occasional positive effects that different ways of associating the pairs $S_{j}$ and $T_{k}$ may have on the registration performance. For example, taking $S_{j}$ and $T_{k}$ in a nonsequential manner would lead to attempted registration in non-contiguous regions of point clouds, what could be guided by the adoption of different principles to span the given axis. Part of the motivation for that is associated with the local properties of the subclouds; if a certain region is not a good candidate for global registration (for example, ambiguity), its vicinity is likely to be a bad choice as well, and hence, some iterations could be saved if the algorithm properly jumped out of that region. The same feeling for non-contiguous spanning can be extended to augment the subclouds by merging non-contiguous slices. These speculations on subclouds composition may require adjustments in block (2) of the pipeline in Figure 3.

Finally, the way the partitioning has been conducted so far can be further investigated, and the algorithm may evolve into a multi-axial partitioning strategy (MPS), meaning that simultaneous cut-sectioning along the three principal axes may be considered (and not along one alone). In this hypothetical approach, the subclouds can be formed by combining the partitions along the three orientations and, ultimately, the concept could be extended to consider other generically oriented axes. This suggestion would, in turn, require adaptations in block (1) of the pipeline shown n Figure 3.

For final remarks, it is important to mention that the existence of a tradeoff between the amount of subclouds and the running time was out of the scope here. In addition, the interesting problem concerning the size of the input clouds and how it can be used to automatically determine a reasonable number of subclouds has been neglected so far. This is an important issue, since the essence of the rationale behind the partitioning concept is that the generated subclouds can individually represent the full clouds, thus allowing for efficient rigid transformation retrieval. By putting these issues as optimization problems and solving them, it is expected that useful insights about the existence and identification of regions-of-interest able to favor global registration of arbitrary scenes will come to light.

## 8. Conclusions

In this work, we introduced an approach for point cloud registration relying on subcloud space, that is, one containing partitions of the original 3D models. Its use along with ICP algorithm was thoroughly investigated based on extensive experiments. The proposed technique drives the conventional ICP into a new use because an outer level of iterations is considered.

A number of outcomes can be drawn:the outer level of iterations favours the correspondence step of ICP and reduces computation efforts: this is because k-registration steps of ($\frac{N}{k}$)-sized point clouds take less time than one registration of N-sized clouds.The existence of two operating modes provides flexibility to the approach, widening the range of possible applications: configuration A is adequate for situations in which little or no information about the scene is provided, as can be the case of huge disorientation between target and source and/or arbitrarily disoriented samples, whereas configuration B suits non-severe disorientation scenario and high-overlapping samples, as can be the case of applications assisted by progressive scene acquisition.The stop criterion based on the micromisalignment concept introduced here performed well, showed to be a reliable measure of quantitative assessment of registration goodness and it is one major contribution of this study to the scientific community.In terms of time performance, comparative analysis revealed impressive results in favour of UPS: except for a few cases in which it was beaten by ICP point-to-plane, UPS was always faster than the other approaches by about 3 times at least. In some cases, it was 300 times faster.In terms of registration quality, UPS performed better than many of the counterparts. In this regard, using RMSE as a metrics for registration quality, UPS was 8 times better than 3D-NDT and GICP in outdoor scenario and 10 times better than Sparse-ICP, Go-ICP and FPFH + ICP in a study of robustness to Gaussian noise.

Summing up, the results obtained surpassed many of the most commonly used registration techniques, as evidenced by a consistent variety of experiments in a wide range of scenarios. The diversity of the investigated scenarios, whether in quantity or at the level of challenge imposed by the scenes, as well as the comparison of the algorithm to counterparts that are recent and relevant according to the literature were useful to demonstrate the generalization of the proposed method.

From the performance analysis in time and quality of pose correction in the different experiments, it can be stated that UPS is a flexible choice for use in robotics and 3D computer vision applications, because it adapted well to the huge variety of scenes, from simple pairwise registration to more challenging outdoor matching of clouds with significant differences in size and overlap ratio.

In conclusion, the algorithm was able to circumvent typical limitations of traditional ICP, such as the subjection to ambiguous registration and the falling into local minima, yet offering remarkable time performance compared to the literature counterparts.

## Figures and Tables

**Figure 1 sensors-22-02887-f001:**
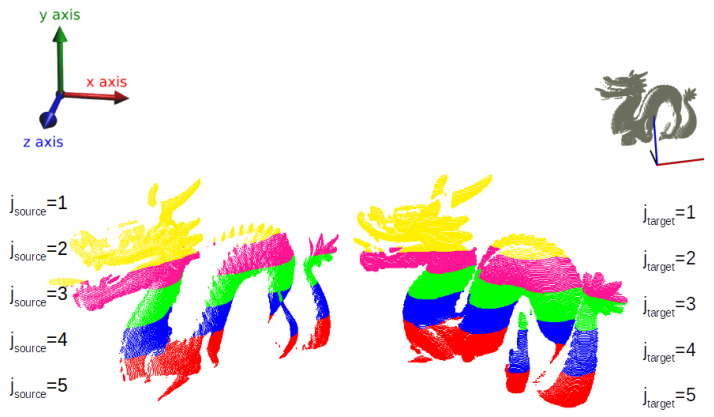
View of top-down partitioning on Dragon model.

**Figure 2 sensors-22-02887-f002:**
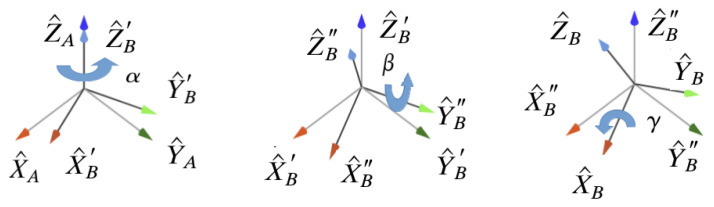
Representation of a sequence of small rotations around principal axes.

**Figure 3 sensors-22-02887-f003:**
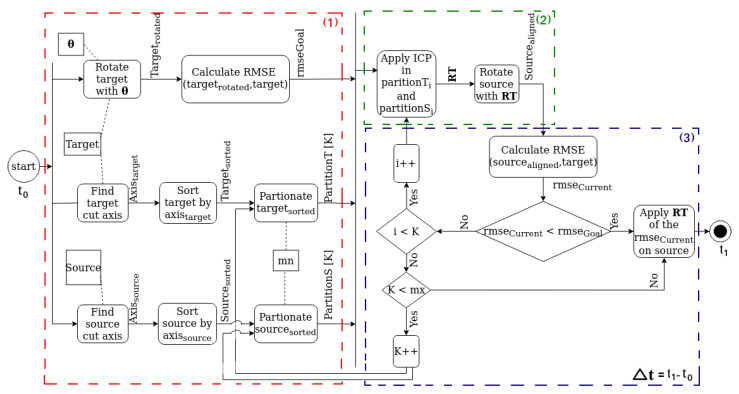
UPS pipeline in three subtasks (red, green and blue colors): (**1**) Partitioning; (**2**) Alignment of the subcloud pairs and (**3**) Quality check of alignment and replication of the local rigid transformation to the entire cloud. Parameter Δt represents the running time in the entire execution of the algorithm and is the quantity present in the tables reported later in Section 6. Finally, mn and mx are, respectively, the minimum and the maximum number of points per partition.

**Figure 4 sensors-22-02887-f004:**
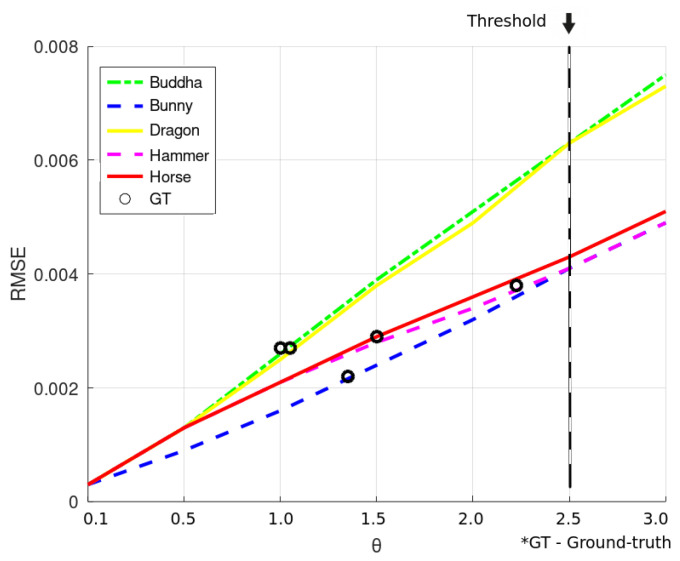
Measurement of the angle value (in radians) to calculate the micromisalignment.

**Figure 5 sensors-22-02887-f005:**
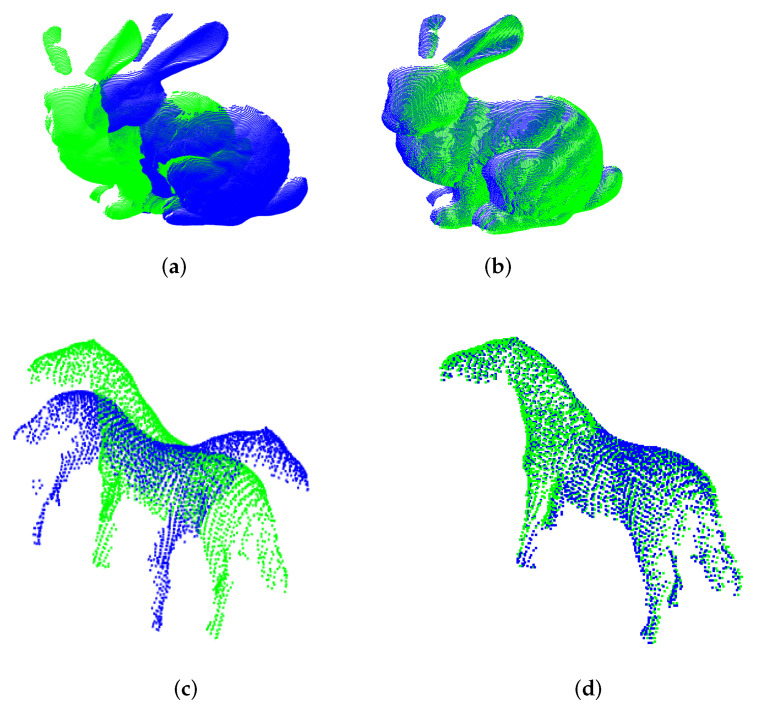
Alignment of the Bunny and Horse models, respectively, for CP-ICP and the proposed method. (**a**) CP-ICP (Bunny model). (**b**) UPS (Bunny model). (**c**) CP-ICP (Horse model). (**d**) UPS (Horse model).

**Figure 6 sensors-22-02887-f006:**
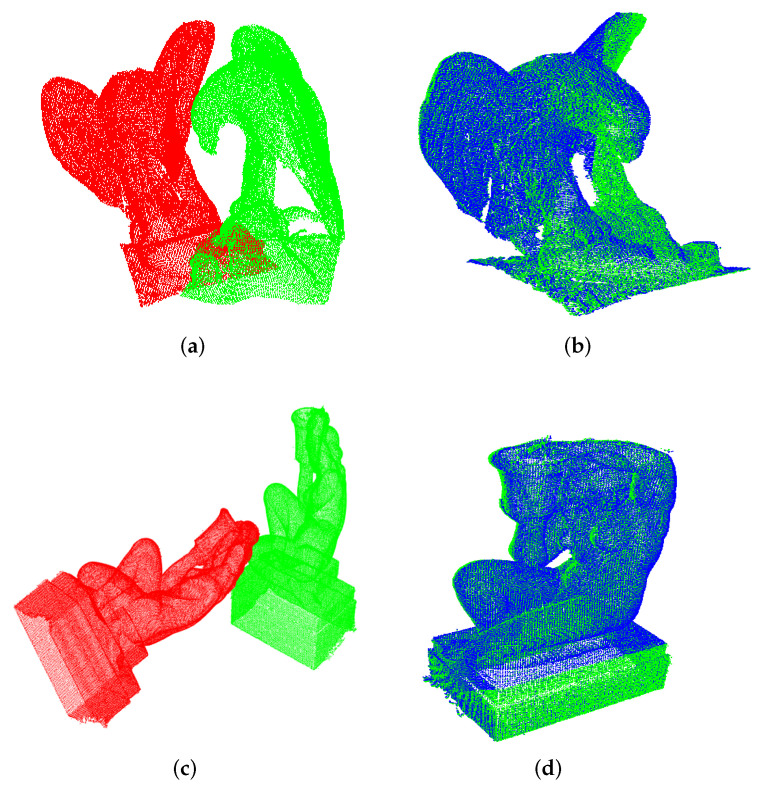
Registration of partial views of the Eagle and Aquarius models under rotational phenomena on generic axes. (**a**) Initial pose (Eagle). (**b**) UPS (Eagle). (**c**) Initial pose (Aquarius). (**d**) UPS (Aquarius).

**Figure 7 sensors-22-02887-f007:**
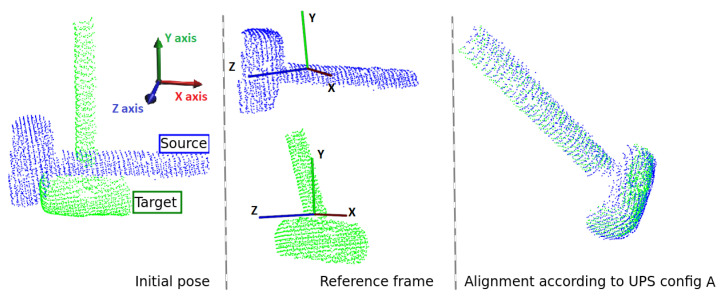
Registration of a cylinder-like object aligned by UPS configuration A.

**Figure 8 sensors-22-02887-f008:**
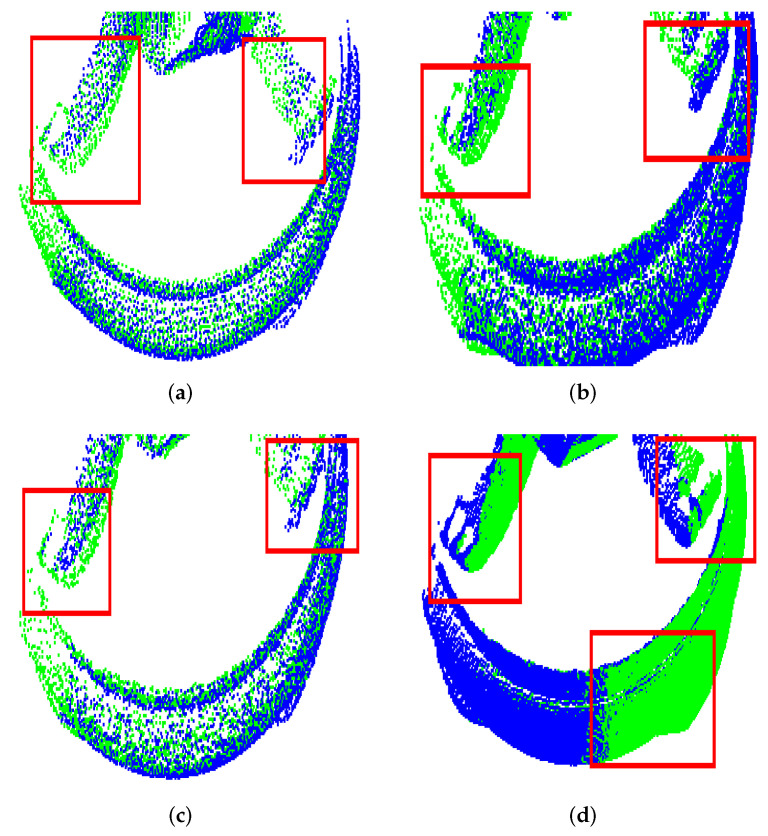
Comparison of the registration results for the Buddha model. (**a**) Uniform. (**b**) Random 50%. (**c**) Random 70%. (**d**) UPS.

**Figure 9 sensors-22-02887-f009:**
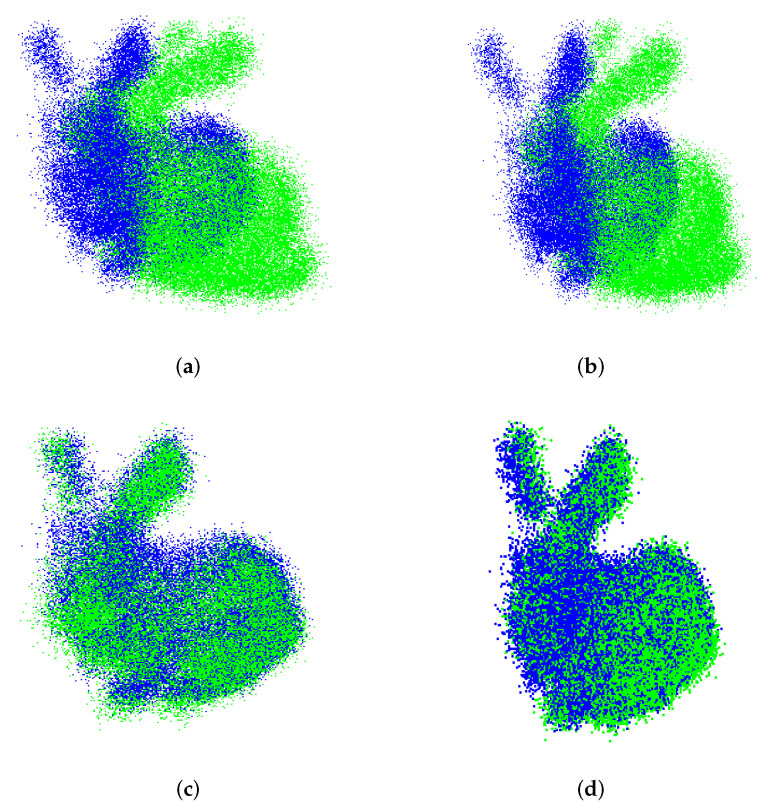
Alignment of Bunny model with added noise. (**a**) Go-ICP. (**b**) Go-ICP Trim. (**c**) FPFH approach. (**d**) UPS.

**Figure 10 sensors-22-02887-f010:**
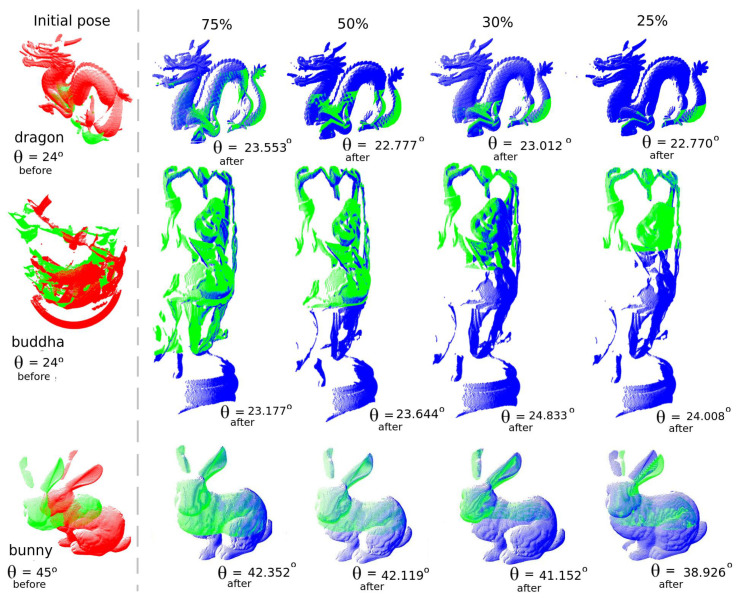
Partial registration of point clouds for the dragon, buddha and bunny models considering 25%, 30%, 50% and 75% overlap ratios. Note in the first column the value of $\theta_{before}$, referring to ground-truth. For the other columns, $\theta_{after}$ means the result of the pose correction.

**Figure 11 sensors-22-02887-f011:**
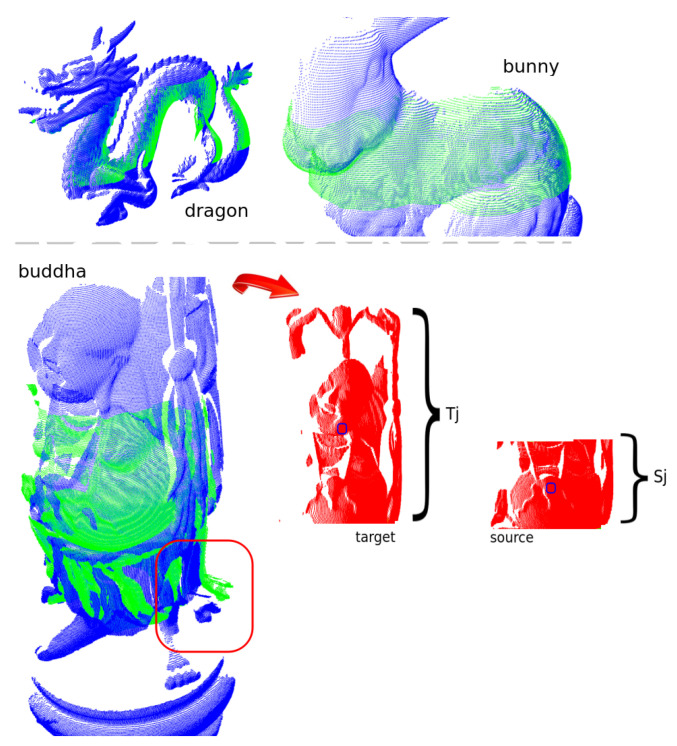
UPS registration corresponding to the case of 50% partial-overlap between source (green colour) and target (blue colour) for the region around the buddha stomach. In red colour, the target and source subclouds selected for the global registration after UPS execution.

**Figure 12 sensors-22-02887-f012:**
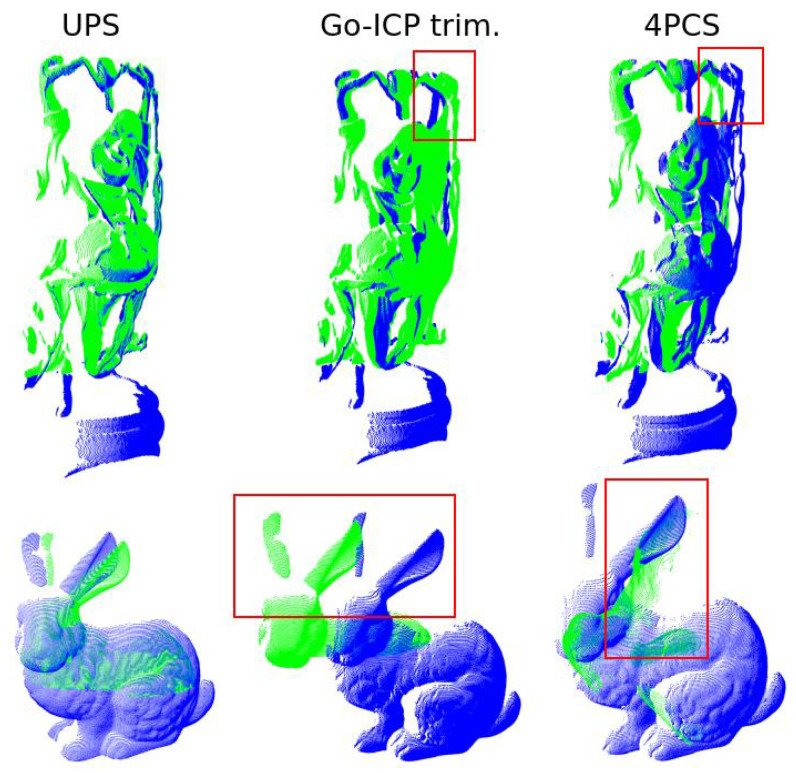
Partial-overlap UPS registration of Buddha and Bunny models and comparison to Go-ICP trimming and 4PCS. In the red boxes, regions with large correction errors are highlighted.

**Figure 13 sensors-22-02887-f013:**
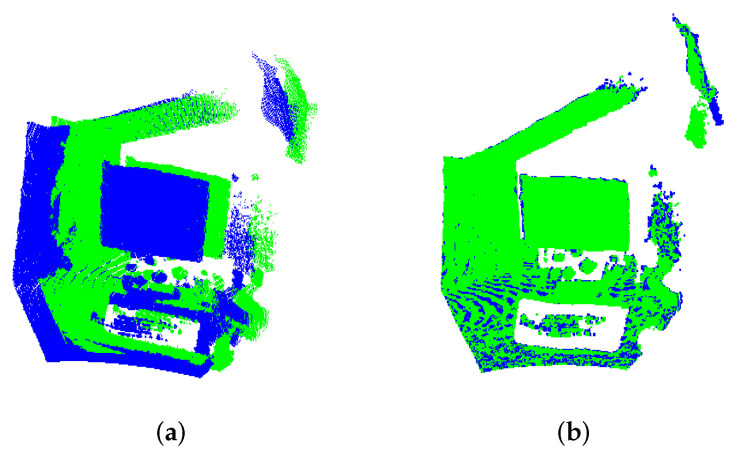
Alignment of indoor scenes of Lab. 1 model: in (**a**) 3D-NDT and (**b**) UPS result.

**Figure 14 sensors-22-02887-f014:**
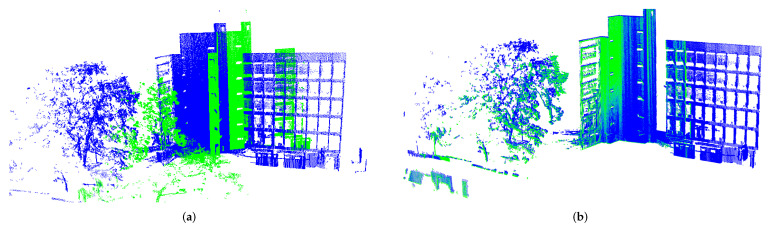
UFC scene alignment under different viewpoints and densities. (**a**) Generalized ICP. (**b**) UPS.

**Figure 15 sensors-22-02887-f015:**
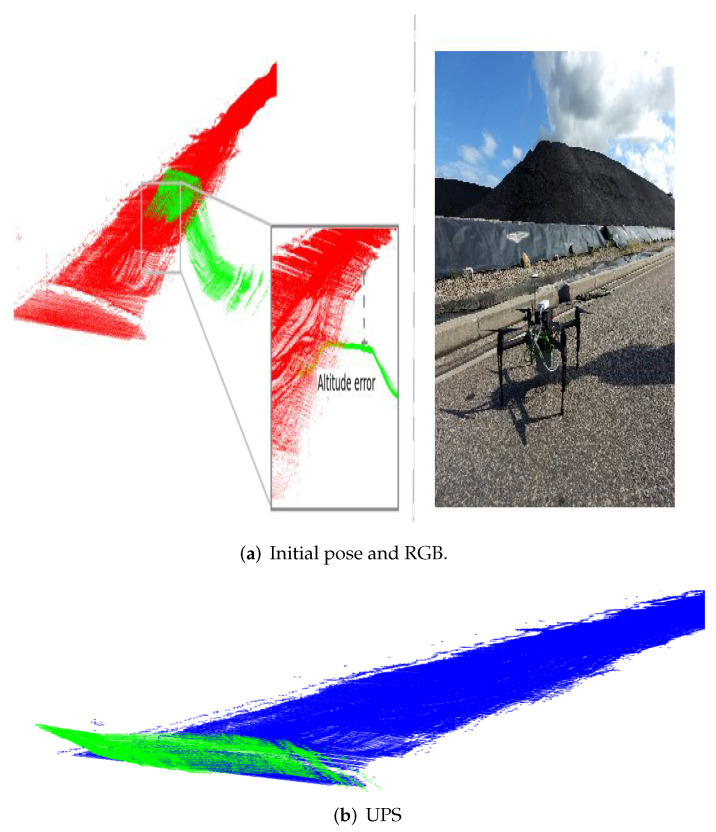
Stockpile scene bad-alignment under different densities and low-overlaping. In (**a**) initial pose and RGB image and (**b**) alignment result.

**Table 1 sensors-22-02887-t001:** Information about the point clouds used in the experiments.

Objects				Scenes			
Model	Dataset	Density	Size	Model	Dataset	Density	Size
Bunny	[48]	40 k	644.3 kB	Lab. 1	Ours	56 k	683.9 kB
Dragon		35 k	1.2 MB	Lab. 2		72 k	879.9 kB
Buddha		75 k	2.2 MB	Office	[23]	200 k	3.6 MB
Horse	[46]	3 k	98.3 kB	Stage		69 k	694.2 kB
Hammer		2k	74.2 kB	House	[39]	83 k	466.1 kB
Aquarius	[47]	64 k	784.8 kB	Gasebo 1	[11]	153 k/67 k	5.1 MB
Bear		27 k	328.5 kB	Gasebo 2		155 k/66 k	5.2 MB
Eagle		68 k	836.2 kB	UFC	Ours	1.2 M/828 k	9.8 MB

**Table 2 sensors-22-02887-t002:** Running time to align pairs of clouds under known ground truths.

	Bunny	Dragon	Buddha	Horse	Hammer
CP-ICP	4.922	3.847	9.085	**0.338**	**0.202**
Go-ICP	36.537	35.847	36.288	42.348	36. 198
FPFH+ $ICP_{p2pt}$	125.438	91.969	413.457	5.526	3.086
$SparseICP_{30}$	22.543	23.260	53.506	2.628	1.454
$SparseICP_{100}$	72.726	55.799	179.481	7.486	4.227
$ICP_{p2pt}$	8.202	6.903	15.585	0.696	0.351
$UPS_{cf.A}$	16.392	3.584	2.501	20.928	1.543
$UPS_{cf.B}$	**3.996**	**1.755**	**2.416**	7.015	0.535

**Table 3 sensors-22-02887-t003:** Rotation obtained from the point cloud registration under known ground truth.

	Bunny	Dragon	Buddha	Horse	Hammer
Ground Truth	45${}^{\circ }$	24${}^{\circ }$	24${}^{\circ }$	180${}^{\circ }$	45${}^{\circ }$
Axis	Y	Z	Z	Z	Z
CP-ICP	16.498${}^{\circ }$	**24.009** ${}^{\circ }$	22.543${}^{\circ }$	55.869${}^{\circ }$	35.075${}^{\circ }$
Go-ICP	34.480${}^{\circ }$	61.281${}^{\circ }$	15.612${}^{\circ }$	42.348${}^{\circ }$	36.198${}^{\circ }$
FPFH+ $ICP_{p2pt}$	46.706${}^{\circ }$	49.605${}^{\circ }$	50.154${}^{\circ }$	187.515${}^{\circ }$	58.929${}^{\circ }$
$ICP_{p2pt}$	41.301${}^{\circ }$	23.863${}^{\circ }$	21.679${}^{\circ }$	36.342${}^{\circ }$	45.577${}^{\circ }$
$UPS_{cf.B}$	**43.246** ${}^{\circ }$	24.091${}^{\circ }$	**24.039** ${}^{\circ }$	**182.610** ${}^{\circ }$	**44.591** ${}^{\circ }$
$UPS_{cf.A}$	**43.246** ${}^{\circ }$	24.091${}^{\circ }$	**24.039** ${}^{\circ }$	**182.610** ${}^{\circ }$	**44.591** ${}^{\circ }$

**Table 4 sensors-22-02887-t004:** RMSE measure after pose correction under known ground truths (in meters).

	Bunny	Dragon	Buddha	Horse	Hammer	RMSE*avg*
CP-ICP	0.011	**0.002**	**0.003**	0.029	0.010	0.011
Go-ICP	0.089	0.055	0.032	0.523	0.207	0.181
FPFH+ $ICP_{p2pt}$	0.004	0.003	0.004	0.004	0.005	0.004
$SparseICP_{30}$	0.057	**0.002**	**0.003**	0.027	0.011	0.020
$SparseICP_{100}$	0.054	**0.002**	**0.003**	0.026	0.006	0.018
$ICP_{p2pt}$	**0.002**	**0.002**	**0.003**	0.020	**0.004**	0.006
$UPS_{cf.B}$	**0.002**	0.003	**0.003**	**0.003**	**0.004**	**0.003**
$UPS_{cf.A}$	**0.002**	0.003	**0.003**	**0.003**	**0.004**	**0.003**

**Table 5 sensors-22-02887-t005:** Elapsed time (in seconds) and RMSE (in meters) of the registration under conditions of rotation on arbitrary axes.

	Aquarius		Bears		Eagle		–
	Time	RMSE	Time	RMSE	Time	RMSE	RMSE*avg*
Go-ICP	55.806	0.714	64.914	0.831	60.492	0.867	0.804
Sparse ICP	798.121	0.802	252.054	1.737	650.190	0.902	1.147
$UPS_{cf.A}$	29.034	0.016	15.400	0.051	2.154	0.034	0.034
$UPS_{cf.B}$	**5.530**	**0.016**	**5.024**	**0.051**	**2.140**	**0.034**	**0.034**

**Table 6 sensors-22-02887-t006:** Effect of downsampling in the registration. View elapsed time (in seconds) and RMSE (in meters).

	*Dragon*			*Buddha*			
	Time	RMSE	θ	Time	RMSE	θ	RMSE*avg*
Uniform	**0.298**	0.004	23.683	**0.462**	0.004	13.184	0.004
Rnd.50% + ICP	5.058	**0.002**	23.873	8.428	**0.003**	21.765	0.002
Rnd.70% + ICP	2.268	**0.002**	23.843	5.140	**0.003**	21.490	0.002
$UPS_{cf.A}$	3.584	0.003	**24.091**	2.501	**0.003**	**24.039**	0.003
$UPS_{cf.B}$	1.755	0.003	**24.091**	2.416	**0.003**	**24.039**	0.003

**Table 7 sensors-22-02887-t007:** Running times of registration methods for noise-influence study.

σ Range	0.002	0.0025	0.003	0.005
Go-ICP	36.811	36.996	36.736	36.707
$Go-ICP_{Trimming}$	37.867	37.389	38.721	37.888
FPFH + $ICP_{p2pt}$	126.108	125.918	125.233	121.885
$SparseICP_{30}$	28.982	29.883	28.535	32.443
$SparseICP_{100}$	96.184	103.004	95.748	105.784
$UPS_{cf.A}$	47.046	48.915	43.434	67.582
$UPS_{cf.B}$	**14.155**	**15.501**	**13.751**	**22.195**

**Table 8 sensors-22-02887-t008:** RMSE (in meters) achieved by registration algorithms on Bunny model with noise level.

σ Range	0.002	0.0025	0.003	0.005	RMSE*avg*
Go-ICP	0.041	0.041	0.046	0.058	0.046
$Go-ICP_{Trimming}$	0.038	0.036	0.041	0.047	0.040
FPFH + $ICP_{p2pt}$	0.016	0.016	0.022	0.016	0.017
$SparseICP_{30}$	0.059	0.057	0.073	0.011	0.050
$SparseICP_{100}$	0.027	0.010	0.062	0.005	0.026
$UPS_{cf.A}$	**0.002**	**0.002**	**0.002**	**0.002**	**0.002**
$UPS_{cf.B}$	**0.002**	**0.002**	**0.002**	**0.002**	**0.002**

**Table 9 sensors-22-02887-t009:** Running time (in seconds) to align pairs of indoor scenes (in seconds).

	Lab. 1	Lab. 2	Office	Stage
$ICP_{p2pln}$	**2.331**	3.986	18.793	21.117
Generalized ICP	12.634	20.897	590.206	397.375
3D-NDT	70.781	138.931	426.403	472.563
$UPS_{cf.A}$	29.649	3.217	78.294	10.908
$UPS_{cf.B}$	5.163	**2.491**	**17.576**	**9.861**

**Table 10 sensors-22-02887-t010:** RMSE (in meters) of registration of pairs of indoor scenes.

	Lab. 1	Lab. 2	Office	Stage	RMSE*avg*
$ICP_{p2pln}$	**0.012**	0.019	0.059	**0.045**	0.034
Generalized ICP	0.024	0.038	0.310	0.175	0.067
3D-NDT	0.046	0.047	0.311	0.167	0.143
$UPS_{cf.A}$	**0.012**	**0.019**	**0.049**	0.047	0.032
$UPS_{cf.B}$	**0.012**	**0.019**	**0.049**	0.047	0.032

**Table 11 sensors-22-02887-t011:** Running time (in seconds) and RMSE (in meters) to align multiple partial shots of indoor scenes.

	Lab. 1		Lab. 2		House		
	Time	RMSE	Time	RMSE	Time	RMSE	RMSE*avg*
$ICP_{p2pl}$	26.209	**0.010**	33.857	**0.010**	40.108	**0.031**	0.017
GICP	116.53	**0.010**	139.341	**0.010**	992.158	0.053	0.024
3D-NDT	131.26	0.054	289.184	0.013	905.530	0.054	0.040
$UPS_{cf.A}$	47.075	**0.010**	5.532	**0.010**	85.812	0.035	0.018
$UPS_{cf.B}$	**10.321**	**0.010**	**5.434**	**0.010**	**20.792**	0.035	0.018

**Table 12 sensors-22-02887-t012:** Point cloud registration in the case of different cloud size.iew elapsed time (in seconds) and RMSE (in meters).

	Gazebo 1		Gazebo 2		UFC		
	153 k → 67 k		155 k→ 66 k		1.2 M → 828 k		
	Time	RMSE	Time	RMSE	Time	RMSE	RMSE*avg*
$ICP_{p2pl}$	6.163	0.248	5.802	0.155	233.987	0.766	0.390
GICP	128.213	0.368	182.603	0.247	2980.82	4.183	1.599
3D-NDT	166.381	0.323	218.221	0.217	256.105	4.300	1.613
$UPS_{cf.A}$	5.501	**0.200**	5.584	**0.147**	545.986	**0.266**	**0.204**
$UPS_{cf.B}$	**3.369**	**0.200**	**3.343**	**0.147**	**60.909**	**0.266**	**0.204**

**Table 13 sensors-22-02887-t013:** Qualitative comparison of algorithms for point clouds registration. Note: *n.a.* stands for not applicable.

	ICP [13,18,19]	NDT [21]	4PCS [51]	FPFH app. [52]	Go-ICP [6]	CICP [40]	Wang [50]	UPS (cf. A)
**(1)** Independent of prior information	×	×	×	✓	✓	✓	✓	✓
**(2)** Independent of coarse-alignment	×	×	n.a.	n.a.	✓	✓	×	✓
**(3)** No need for sampling	×	×	×	×	×	✓	✓	✓
**(4)** No performs of registration in feature space	✓	✓	×	×	✓	✓	×	✓
**(5)** Robust to loss of surface details	✓	×	×	×	×	✓	×	✓
**(6)** Multi-scenario scope	×	×	×	×	×	×	×	✓

## Data Availability

The data presented in this study are openly available in Stanford 3D repository [link https://graphics.stanford.edu/data/3Dscanrep/ (accessed on 15 February 2022)], Statue Model Repository [link https://lgg.epfl.ch/statues_dataset.php (accessed on 15 February 2022)] and Mellado’s dataset in Super4PCS paper [link http://geometry.cs.ucl.ac.uk/projects/2014/super4PCS/ (accessed on 15 February 2022)]. Some data were obtained from Jacopo Aleotti (Horse and Hammer model) and Marcus Forte (UFC and Stockpile dataset) and we cannot make them available.

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
