# Peer review of "Uniaxial Partitioning Strategy for Efficient Point Cloud Registration"

_sensors, 2022, doi:10.3390/s22082887_

Round 1

Reviewer 1 Report

My feeling is that the proposed Uniaxial Partitioning Strategy for Efficient Point Cloud Registration is in principle an ICP under the RANSAC framework. That is, by partitioning the source points cloud and target point cloud into sub-clouds, pairs of sub-source and sub-target cloud is registered by ICP to give a possible rigid transformation, then this transformation is used to verify the original source and target clouds to see whether they are registered satisfactorily.

The manuscript is lengthy, but the key contribution is a short part, that is, the part of the so-called uniaxial cloud partitioning strategy. The underlying rationality is simple: To partition both the source and target clouds into K sub-clouds of the same point size, then use the ICP for registering a pair of smaller-sized sub-clouds to rapidly obtain a possible rigid transformation. It is precisely this strategy that troubled me, in particular, for partial registration. I have two concerns on it, one is a major one, the other is a minor one, and I would seek more clarifications on these two issues.

The authors should clarify how to determine the axis, since it critically affects the effectiveness of the proposed partition strategy. For example, given a vertical cylinder, if the source axis is vertical, but the target axis is horizontal, then the partitioned two sets of K sub-clouds seem difficult, if not impossible to register.  Of course, this is an extreme case. However, how to determine the axis should be properly addressed, in particular, for partial registration. Since both source and target clouds are partitioned into K sub-clouds, suppose source is only a small part of the target, the partitioned sub-cloud registration is equivalent to the partial registration with a smaller overlapped part, hence becoming more challenging. This is my major concern, and I hope the authors should address this issue carefully.

Note that if the axis is known, the source and target clouds registration becomes largely a translation operation, rather than a rotation plus a translation. Hence knowing the axis is in fact a provision of substantial extra information

A minor problem is that, having partitioned both the source and target into K sub-clouds, the pair of source sub-cloud and target sub-cloud with the same index is used for registration. It seems to me it lacks any theoretical justification in partial registration . Of course, source and target could be partitioned into different number of sub-clouds of similar size, and different pairs of source and target sub-clouds could be assessed. This is theoretically more reasonable, but more computations are needed, which in turn offsets the computational advantages of the proposed strategy.

Anyway, I would authors carefully address these two issues before making any recommendations.

Other points

The rigid transformation is not a linear transformation

Introduction should be substantially reduced. Those work not closely related this one should be removed;

Please focus more on partial registration

Author Response

Dear Referee #1,

First of all, many thanks for your valuable and deep revision of our manuscript, which we found very useful for its improvement. After careful discussion of your major and minor concerns amongst us authors, we were finally able to justify and answer them all as it follows.

======

  1. You say "The manuscript is lengthy..." and later you suggest removing citations to topics other than partial registration “Those work not closely related to this one should be removed”, and to correct the mention of “linear transformation” (line 24 of the introduction).

reply: since the paper length was a concern of another referee in the present submission, we authors decided to change the introduction by canceling any mentions to pose and gesture estimation, as well as to the problem of non-rigid registration. Additionally, we also canceled one full paragraph (lines 43 to 47) on feature space registration, since ours is not in that space, but in spatial-coordinates space. Finally, we canceled the word “linear” in line 24.

======

  1. You show concern on the performance in case of partial registration: "It is precisely this strategy that troubled me, in particular, for partial registration"

reply: In this regard, we must say that your concern is fully relevant; as a matter of fact, for ICP-like registration techniques, it is well-known that correspondence between target and source is a must. To be stressed that this has been also our concern from the very beginning; that is why we had already included a section focused on the problem of partial overlapping, in which we investigate the performance of our strategy and compare it to others. In that investigation (see the content of section 6.5 for complete discussion), we faced the problem of overlapping ratios ranging from 75% to the very challenging case of 25%, thus allowing us to assess the limits of our techniques (i.e., when it starts failing) and, despite the fact that UPS registration does not get stuck, the retrieved orientation does deviate from the correspondent ground-truth of Dragon and Bunny models. Summing up, we do not claim that our approach is able to provide matching under quasi-null overlapping scenarios, but we firmly believe that it performed good enough as compared to other approaches (see Figure 12 for qualitative comparison to Go-ICP and 4PCS).

Furthermore, in that same section we moved forward and analyzed also the role of the specific portion in which overlapping between source and target takes place. In the discussion coming along with the results, we made important comments and highlights. For example, in Figure 11 we showed how the size difference in target and source slices as well as the relative displacement of their centroids led to bad registration of the medium portion of the Buddha model (around its stomach).

======

  1. You suggest an interesting and extreme case study as you note a lack of comprehension on the choice of cutting-axis “The authors should clarify how to determine the axis, since it critically affects the effectiveness. For example, given a vertical cylinder…”:

reply: In this regard, we must first say that we have been aware of the importance of the choice of the cutting-axis so far, but the reviewer really alerted us to the need of a new experiment to help judge about the strategy effectiveness and to better illustrate the issue.

For what concerns the choice of the cutting-axis, we had initially explained about the existence of configurations A and B, or, in other words two possible operating modes (see section 3 and  lines from 230 to 236) and then added, in lines from 257 to 267 important information regarding how the axes are chosen and also applications in which each of the configurations would suit better (according to our feeling). For brevity, essential for the choice of the cutting-axis is the calculation of the data dispersion in the three principal axes of the local frame (x, y and z) in which the point cloud data is provided after acquisition. For completeness, we provided an additional paragraph (see lines 237 to 243) to justify why not relying on the calculation of PCA. 

Reading carefully the reviewer’s comments when he says “...then the partitioned two sets of K sub-clouds seem difficult, if not impossible to register”, we must say that, for the mentioned case of huge source-target disorientation,  the real problem of the mentioned lack of correspondence between sub-clouds appears in Configuration B, but not in Configuration A. That is because in Configuration A, the algorithm is free to choose cutting along the x-, y- or z-axis of the models. The information on data dispersion used to set the cutting-axis is, therefore, a provision of useful information to help the algorithm understand the pose of source and target. Note that, with that piece of information about the topology of the object-of-interest, the cutting-axes of source and target can be different, but still the slices will contain correspondent (or overlapping) samples of the object surface, which was confirmed by the numerous tests reported in the manuscript.

To better illustrate the issue, we considered the important case study suggested by the reviewer and added it to the present version of the manuscript in section 6.2, lines 437 to 448. The cylinder-like object we had available is the Hammer object of prof. Aleotti’s database (Parma University, Italy), see Figure 7 in the paper. We then considered initial poses as challenging as the reviewer suggested and used them as inputs to UPS. The algorithm worked in Configuration A, cutting along the Y-axis the source model and the Z-axis the target model with 4 slices of nearly 500 points each. After 1.05664 seconds, the algorithm met the stop criterion quality and reached RMSE=0.00407211. Although UPS running in Configuration A was able to handle that extreme scenario, we think that other new configurations can succeed, especially if other ways of selecting subcloud pairs and composing segments are available, as mentioned in the discussion in lines 657 through 672. 

=====

  1. You noticed that an indexation correspondence of source and target slices has been considered so far, but suggest that they could be evaluated in a free-association way “...and different pairs of source and target sub-clouds could be assessed”

reply: We completely agree with the reviewer that the use of different associations of source and target slices could lead to a more general search space of candidates to match and, as such, better outcomes could be achieved. In that augmented space, one interesting initial investigation to point attention is on close (but non-contiguous) sub-clouds, since the proximity helps preserve the correspondence of surface points. Our guess is that, however, that would come at the cost of rising computing efforts (which has been one of our main goals so far). In this context, one could also get inspired by the use of optimization algorithms to propose new uses and versions of the UPS introduced here. Those aspects were highlighted now by the text added in lines 550-557 and were already mentioned in the first version in lines 180-182 .

For what concerns the lack of what the reviewer called “theoretical justification” for choosing the index-associated pairwise registration in the present version of the technique, we would like to mention that the information given previously about the way the cutting-axis is chosen may suffice. Indeed, since the slicing in both source and target models take place along correspondent - though partially overlapped- regions, the first natural attempt is to check slices sequentially. This is what we have done so far and it has yielded good results, though we know that aggressive rotations (eg 180°) may represent a limitation for our approach, as the correspondence would be concentrated in the central slices

Finally, we would like to stress that the multitude of experiments in different scenarios and for different data characteristics somehow works as a good piece of generalization of the strategy introduced, confirming its effectiveness and, in our point-of-view, alleviating the need of deeper theoretical rationale.

=====

Reviewer 2 Report

This paper proposes an improvement over a previous conference work which enhances the traditional ICP method for registration purposes. The paper is well written and structured. Results are tested against standard methods available, amongst other, well-recognized libraries such as PCL. Notwithstanding that point cloud dataset are synthetic for the main comparative tests, the results are valid, and there is also a final section where a real dataset is used. This fact supports its validation.

I suggest the the synthetic point cloud datasets are introduced with more details: resolution, size, file extension, etc.

Author Response

Dear Referee #2,

First of all, many thanks for your valuable revision of our manuscript, especially for the compliments and positive aspects identified in the methodology. After careful reading of your recommendations, we proceeded to change the manuscript so as to include additional information regarding the point cloud models used in the study. In doing that, a new Table (Table 1) has been added on page 9; it brings the names, the repository where each one can be retrieved, the amount of points (which gives an insight on how dense the point clouds are), as well as the corresponding file size separately for objects and scenarios. We emphasize that all the models used so far are inside archives of .pcd extension type, and that is why we put that piece of information in line 344 and omitted it in the table. We hope the current version meets the required modifications.

Reviewer 3 Report

Dear Authors

The paper topic is interesting in terms of new mathematical development. Although it would possess rational material to be considered as a potential for publication, however, The Reviewer is recommending a revision of this article at this format.

Nevertheless, the reviewer recommends the Authors to carefully address the following comments in the revised version.  

COMMENTS

  1. What do you mean of RMSE in the abstract? It is not clear for the readers.
  2. What is the novelty of the studied work? What is its main contributions to the state of the art? It is not clear in the text.
  3. The language must be improved. The Reviewer strongly recommends the Authors to review the English. For instance, all the active tenses starting with “WE” must be replaced by the passive tense, more academic.
  4. Why did you consider too many subsections? It is not essential to divide the paper in such subsections. Please correct it in the revised version.
  5. Figure 1 is the analysed model of the dragon, is this corresponding to a real model?
  6. The information regarding software that the authors used to develop the results is missing.
  7. How the Threshold equation was derived, Equation (11)?
  8. The density of the obtained point clouds is not clearly mentioned in the manuscript. Please clarify this.
  9. What is the solution accuracy of the developed methodology?
  10. Please rewrite the conclusions. It must support the main outcome of the carried out study. It is recommended to present some bullets.

Very Best

The Reviewer

Author Response

Dear Referee #3,

First of all, many thanks for your valuable and deep revision of our manuscript, which we found very useful for its improvement. After careful discussion of every question and concern that came out in your revision, we were finally able to justify and answer them all as follows.

======

  1. You asked about RMSE and suggest that the reader may not comprehend it

reply: The acronym stands for Root Mean Square Error, a common metrics for measuring deviation among samples of multidimensional data. In the present paper, we used it to quantitative evaluate the quality of registration, as we mention in line 396 for example: “Quantitative assessment is based on RMSE…”

Nevertheless, from your question we could detect that in the first appearance of the acronym, in the abstract, the reader could miss that information, and therefore we modified it to include the acronym definition inside brackets.

=====

  1. Your question is about the novelty and contributions of the present paper compared to the state-of-the-art; you suggest to write it down clearly in the paper

reply: The paper has a whole section for that (please refer to section 3), but we added new text to make it more clear for readers. The new text appears in blue color in the present version of the manuscript and it highlights the innovation and versatility brought by the possibility of two operating modes (Configurations A and B). The new text also emphasizes the contribution we offered to the scientific community by the introduction of a micromisalignment measure as a new stop criterion for the registration. 

In addition, it should be mentioned that in the section we also discuss the remaining improvements of the present version of the technique compared to previously published ones (for example, the way the cutting-axis is chosen). Finally, in the last paragraph of that section we discuss conceptually how the introduced approach is placed in the literature; essentially, we explains that it does not work in feature-space nor does in sampled space, but it is instead a geometry-preserving approach in the sense that the acquired surface itself undergoes registration in spatial coordinates space.

=====

  1. You recommend an English language review and points out for the excessive use of active tense

reply: In this regard, we must say that several occurrences of active tense were corrected and replaced by the use of passive tense. Every correction is shown in blue color alongside the older text having the strikethrough resource activated.

Furthermore, we stress that the current version of the manuscript went through an automatic correction in the following platform https://preflight.paperpal.com/partner/ieee/access.

=====

  1. You showed concern about the number of sections and recommended to reduce it

reply: We agree with the referee that there are many sections; however, we have reasons for that. First of all, we made that choice for organization purposes: since we investigate a wide range of experiments which differ a lot in many aspects, we preferred to have many sections and to present in detail the conditions set for each one inside them.

Indeed, the plural investigation reported in the paper addresses investigation in:

- the number of samples (if pairwise registration or multiple shots); 

- the size of clouds (if sampled or dense data, and also if unbalanced in size); 

- in the acquisition conditions (regarding the existence of noise, the role of overlapping);  

- in the origin of data (if objects or indoor or still outdoor scenario). 

In addition to that, we also think that any modification in the original structure of the paper could compromise the progressive connection and cohesion of the text among sections, which ultimately could affect the comprehension of it and its readability.

Based on that, we respectfully preferred to keep the organization of the paper as submitted.

=====

  1. You asked about Figure 1, whether it corresponds to a real model.

reply: That is a 3D model of a dragon statue as acquired by the 3D scanner Cyberware 3030 Model Shop, and made available for point cloud researchers through a public repository of the Stanford University. To make the art of Figure 1, which is naturally a 2D projection of the point cloud, we divided the cloud into five slices and changed the color of the points lying in the slices so as to ease the visualization of the sub-clouds, thus favoring the comprehension of our uniaxial partitioning strategy.

=====

  1. You mentioned about the lack of information regarding the developed software codes

reply: We thank the referee once again for the warning. The paper was updated with important information regarding the implementation details in section 5.1. In the lines therein, the reader is informed that all the steps of our UPS were developed in C++, meaning that a large set of codes have been written for the micromisalignment, for the choice of the cutting-axis, for the cutting itself (partitioning) and for the assessment of registration quality. For what concerns the input of clouds and the in-core registration, we resort to the ICP implementation of the Point-Cloud Library (PCL), which is a benchmark framework for this purpose.

It is important to mention that the input parameters of every registration experiment presented in the manuscript as well as the executable program (including those related to the literature and state-of-the-art techniques used for comparison) can be downloaded from https://github.com/pneto29/UPS_paper. In the paper, the reader is made aware of this with the text added in lines from 385 to 388. 

=====

  1. You ask how the micromisalignment measurement (equation 11) was derived.

reply: To comprehend that threshold measurement, it is essential to notice that the micromisalignment is a deviation measurement between two sets of 3D points. In the present work, the metrics used for that deviation is the root mean square error (RMSE). In the computation of RMSE, the term regarding the 2-norm contains the difference between two vectors: one of them denotes a point of the target and the other denotes a 3D spatial-deviation of that vector imposed by the left multiplication by a rotation matrix. The trick is that, for that deviation to be slight or subtle, the rotation matrix should represent a micro disorientation around the three spatial directions (x, y and z). That is exactly the concept of micromisalignment (explained in section 4.3.2) comprised by the rotation matrix of Equation 10 which appears in the formula used as stopping criterion in the UPS.

=====

  1. You observed that we missed the information regarding the point cloud density.

reply: We thank you again for the warning. In this regard, we decided to update the paper to include important information about the point cloud models used in the study. In doing that, a new Table (Table 1) has been added on page 9; it brings the names, the repository where each one can be retrieved, the amount of points (which gives an insight on how dense the point clouds are), as well as the corresponding file size separately for objects and scenarios. We emphasize that all the models used so far are inside archives of .pcd extension type, and that is why we put that piece of information in line 344 and omitted it in the table.

=====

  1. You asked about the solution accuracy of the proposed approach

reply: In the present study, we used as accuracy indicator the RMSE metrics already mentioned. As pointed out earlier (in this response letter) and in the paper itself, the RMSE is commonly used for that purpose in the field of point cloud registration. Since it is strictly dependent on the amount of points of a given cloud sample, it is expected that different registration experiments (running on clouds of different size) yield different results on that figure. Therefore, based solely on RMSE, we are not able to provide an absolute solution accuracy as required, but we can use it as a relative measurement.

To meet that, for some of the experiments we calculated an average of the outcome RMSE obtained for the different samples considered. For example, in section 6.1, Table 4 was updated with a column showing the average RMSE as achieved by UPS and the counterparts as well. The same thing was done  for the remaining experiments of sections 6.1 through 6.8, and the tables presenting RMSE results were updated in the current version of the paper. 

The use of average RMSE revealed that globally UPS performs better than any other approach used for comparison in the present investigation.

=====

  1. You recommend the conclusion to be rewritten and to use bullets for making it easier to see the main outcomes of the study

reply: As required, we provided modifications mainly in the form of the conclusions to highlight the outcomes of the present study. The text added or modified into bullets and listing is now in blue color for clarity.

Round 2

Reviewer 1 Report

The revised text looks better, and the authors  addressed my concerns to a large extent

Author Response

Dear reviewer,
all previous comments have been adopted in the current version. We are currently in a second round of review with the editor, emphasizing the changes previously listed.